

# Flood exposure in Rotterdam's unembanked areas from 1970 to 2150: sensitivities to urban development, sea level rise and adaptation

Cees Oerlemans[1,2], Martine van den Boomen[1,3], Ties Rijcken[1], Matthijs Kok[1]

[1]Faculty of Civil Engineering and Geosciences, Delft University of Technology, Delft, 2628CN, The Netherlands
[2]HKV, Lelystad, 8232JN, The Netherlands
[3]Rotterdam School of Applied Sciences, Rotterdam, 3013AL, The Netherlands

*Correspondence to*: Cees Oerlemans (c.oerlemans@tudelft.nl)

**Abstract.** Uncertainties in the rate of sea level rise, coupled with ongoing urban expansion, create challenges for city planners in designing flood risk adaptation strategies. This study analyzes flood exposure rates in Rotterdam's unembanked areas from 10  1970 to 2150. We modeled flood hazards for 10-1000 year return periods under both low (RCP2.6) and high (RCP8.5) emission scenarios, while assessing exposure using historic and planned urban development data. Without adaptation measures, flood exposure in Rotterdam's unembanked areas is projected to increase. For 10-year flood events under RCP8.5, exposure rates are expected to increase 7-fold by 2150 compared to 2020. For RCP2.6, a 3-fold increase is projected, reflecting uncertainties in long-term sea level rise. A retrospective analysis reveals an improvement in flood exposure: exposure levels observed in 15  2020 were approximately half those observed in 1996, due to construction of the Maeslant storm surge barrier. Temporal variations in exposure rates are attributed to three factors: urban development, sea level rise, and the construction of the Maeslant barrier. Exposure rates are primarily influenced by the Maeslant barrier, followed by sea level rise and urban development. Understanding the interplay of these three factors is crucial for urban planning and flood risk management in delta cities.



## 1 Introduction

Without adaptation, the risk of flooding is expected to increase due to the combined trends and interactions of economic growth and climate change. Over the past decades, the economic impact of flooding events has steadily increased, primarily driven by

socio-economic development in flood-prone areas (Aerts & Botzen, 2011; Jongman et al., 2014). Globally, there is a persistent migration trend toward coastal areas and from rural to urban settings. This leads to higher population density, which in turn heightens coastal exposure and vulnerability (Andreadis et al., 2022).

Climate change exacerbated these risks, manifesting through more frequent storms, rising sea levels and higher river peak

discharges (Calafat et al., 2022; Oppenheimer et al., 2019). Several studies have disentangled the key drivers of flood risk - hazard, exposure, and vulnerability - for both riverine and coastal flooding. Historical analyses from 1950 to 2020 show that the most important drivers of flood impacts in Europe have been exposure growth and vulnerability decline (Praprotny , 2024). Steinhausen et al. (2022) analysed the independent and combined influence of exposure change and climate scenarios on flood risk in Europe. They find that in all future periods – 2025, 2055 and 2085 – exposure has a greater influence risk change than

climate driven hazard. Global analyses reveal distinct regional patterns. While flood impacts in African countries are mainly driven by climate change, in growing Southeast Asian economies (Indus, Yangtze, and Mekong basins), rapid urban growth dominates over climate effects (Winsemius et al., 2016). Koks (2014) provided a framework to jointly assess flood hazard, exposure, and social vulnerability at regional scales, demonstrating that including detailed regional information on flood risk drivers is crucial for developing effective flood reduction strategies. Understanding these different drivers has contributed to

the recognition that adaptation is essential and can significantly reduce future flood losses, even under lower global warming scenarios (Magnan et al., 2022; Oppenheimer et al., 2019). Adaptation strategies consist of measures on various scales, including structural adaptations (e.g. sea walls, levees), nature based-solutions (e.g. mangrove restoration, wetland creation), building level adaptation (e.g., elevating structures, flood-proofing) and future urban development policies (Aerts, 2018; Song et al., 2017).


The implementation of adaptation strategies in urban port cities presents challenges due to limited space and the complex interplay of diverse stakeholder interests. At the same time, relocation of port activities to accommodate larger cargo ships has created new opportunities for urban development in unembanked areas. Examples of port areas where these developments are happening are the city of Houston (Brody et al., 2018), the city of Copenhagen (Hallegatte et al., 2011) and the city of

Rotterdam (de Moel et al., 2014). While these developments offer potential economic benefits, they can also result in more flood exposure, as unembanked areas lie between flood sources and existing flood defence systems (Kaufmann et al., 2018).

      The case of Rotterdam provides a compelling example of the complexities involved in managing flood risks in unembanked areas. As the need for housing is high in the Netherlands, and especially in the larger Rotterdam area, the city is making critical decisions about where and how to build. The (re)development of unembanked areas requires careful

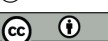



consideration of various trade-offs. For instance, policymakers must weigh the benefits of neighbourhood-level adaptations like ground raising, which primarily benefit the adaptation of new real estate, against more expensive system-scale measures such as strengthening storm surge barriers, which protect both new and existing buildings. These decisions have far-reaching implications for urban planning, flood risk management, and the overall resilience of the city.

Previous flood risk studies that combine and correct flood losses for urban development have been limited in temporal and spatial extent, leading to an incomplete presentation of trends in flood exposure over time (Paprotny et al., 2018). In the Netherlands, studies assessing how flood risk might evolve in Rotterdam's unembanked areas (de Moel et al., 2014; Veerbeek et al., 2010), have focused only on future scenarios, and do not describe historical trends. These studies conclude that more accurate risk estimates would benefit from a more detailed consideration of objects and land-use categories. Moreover, as

urban planning and water management evolve towards longer decision-making timeframes, there is a need for flood risk assessments that can inform these extended planning horizons while capturing temporal path dependencies.

    This research aims to address these gaps by proposing a structured and flexible assessment framework for analyzing historical, present, and future flood exposure in unembanked areas. Our framework is designed to unpack total flood exposure in a manner

that supports decision-making in urban planning and flood adaptation responses. The framework is applied to the unembanked areas of the city of Rotterdam and explicitly distinguishes between the impacts of urban development, sea level rise and adaptation efforts on flood exposure.

    The study follows a four-step process:

1)   Total flood exposure in Rotterdam's unembanked areas is analyzed from 1970 to 2150, considering both low and high emission scenarios. Without additional flood risk mitigation investments, projections indicate a significant increase in exposure rates.

    2)   The study characterizes the spatial variation of flood exposure by evaluating temporal exposure rates across different neighbourhoods. This analysis reveals significant variability in exposure rates, underscoring the importance of

80           designing tailored long-term adaptation strategies for each neighbourhood.

    3)   Exposure rates over time are attributed to three key factors: urban development, sea level rise and the influence of the Maeslant barrier. By clarifying these factors' individual and collective impacts, the analysis provides crucial insights into the primary drivers of flood risk in Rotterdam.

    4)   Various urban development strategies are explored by varying the design flood elevation for new urban development.

85           Results show that while increasing this elevation does mitigate exposure rates, significant reductions are primarily observed for extreme events with return periods of 1000 years or more within the specified timeframe.



The framework is designed to be applicable to other unembanked areas in deltas worldwide. It offers flexibility, allowing models and data to be adjusted based on specific locations and desired temporal and spatial scales.

## 2 Methods and data

Flood risk is commonly defined as a function of flood hazard, exposure and vulnerability. Although general damage drivers for vulnerability modelling are widely recognized, the exact effect of hazard characteristics on exposed structures is still poorly understood and largely depending on the material and its quality (Huijbregts et al., 2014). The present study primarily focusses on the spatial variation in exposure among unembanked neighbourhoods and the attribution of key factors, particularly urban

development, on flood exposure. Hence, we quantify exposed buildings rather than modeling economic values using damage curves. Vulnerability is deliberately excluded from the scope of this study due to the scarcity of detailed empirical data from flood events, which are necessary for calibrating damage curves. Flood damage models are even less frequently calibrated in the Netherlands, where current estimates are based on flood damage records from the coastal flood of 1953 (Slager et al., 2013). A limitation of these flood vulnerability estimates is their generally limited transferability between different flood types

(Wagenaar et al., 2018). This limitation is particularly evident in Rotterdam's unembanked areas, which feature a diverse range of building types, including both low-rise and high-rise structures.

Our research methodology focusses on flood hazard and exposure modelling (Fig. 1). We assess exposed buildings resulting from urban development, sea level rise, and the Maeslant storm surge barrier construction, attributing impacts to each factor.

Six combinations of hazard and exposure are used in this study, as outlined in Table 1 and noted in the scenario legend of Fig. 1. The reference scenario includes sea level rise, urban development, and the construction of the Maeslant barrier. Subsequent scenarios explore exposure rates under various conditions, including scenarios without sea level rise, without urban development, and without the construction of the Maeslant barrier. The final two scenarios can be interpreted as variations on the reference scenario, differentiated by alterations in the design flood elevation. In cases where no design flood elevation is

specified, this indicates the preservation of the current elevation distribution within each neighbourhood.





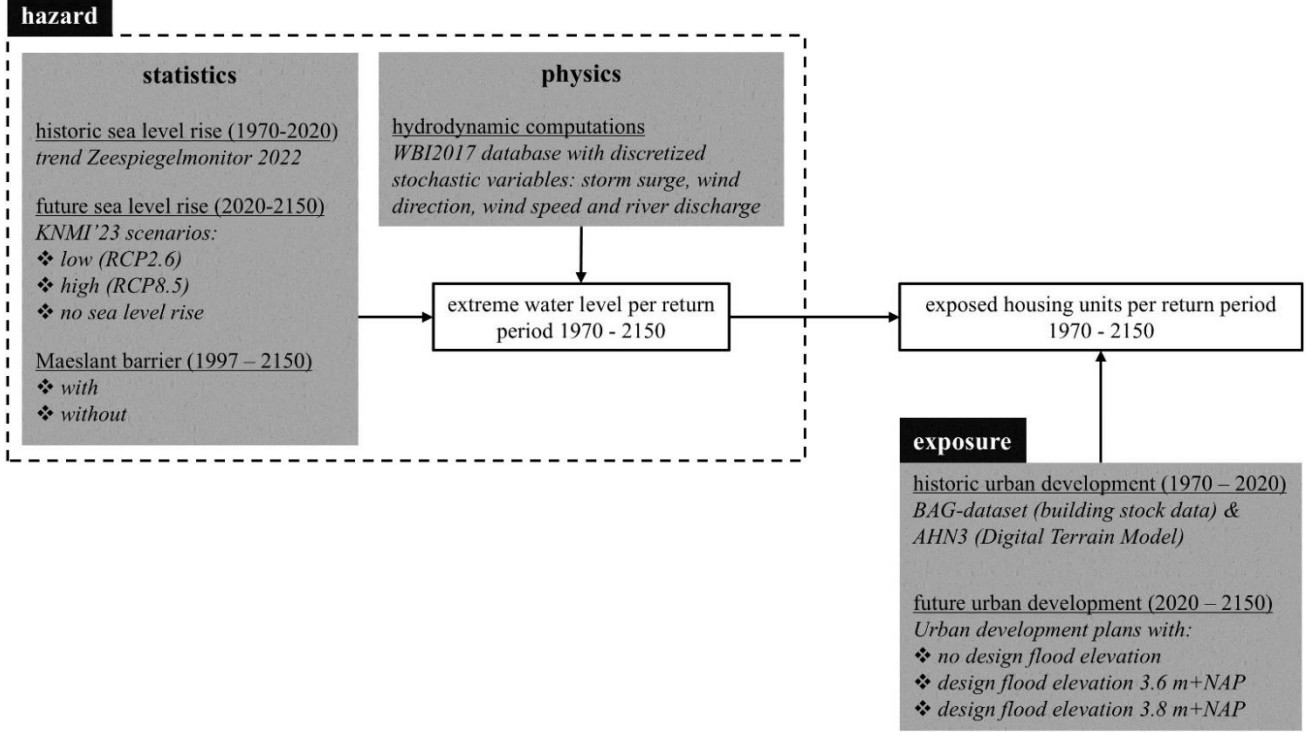

**Figure 1: Methodological framework and data sources for calculating exposed housing units.**

In the remainder of this section, we will outline our approach and the data used. We begin by describing our flood hazard modelling methods in Sect. 2.1, followed by our exposure modelling techniques in Sect. 2.2. Section 2.3 then explains how

we combine hazard and exposure data to create spatial and temporal maps of flood exposure. Throughout these sections, we detail the specific data sources used for our case study in Rotterdam. While our methodology is tailored to the unembanked areas in Rotterdam, it can be adapted to other contexts. Researchers applying this framework to other port areas or urban coastal regions can substitute local data for both the hazard and exposure components, making the methodology broadly applicable.


**Table 2: Scenario-specific information included in the exposure analysis, color-coding matches colors in subsequent figures. NAP = Dutch Ordnance Datum, which approximately corresponds to mean sea level.**

| Scenario | | Design flood elevation | Sea level rise | Urban development | Maeslant barrier |
|---|---|---|---|---|---|
| Reference | ● | NAP +3.6 m | 1970 – 2150 | 1970 – 2040 | 1997 - 2150 |
| No sea level rise after 2020 | ● | NAP +3.6 m | 1970 – 2020 | 1970 – 2040 | 1997 - 2150 |
| No urban development after 2020 | ● | N/A | 1970 – 2150 | 1970 – 2020 | 1997 - 2150 |
| No Maeslant barrier | ● | NAP +3.6 m | 1970 – 2150 | 1970 – 2040 | None |
| No design flood elevation after 2020 | ● | None | 1970 – 2150 | 1970 – 2040 | 1997 - 2150 |
| Raised design flood elevation after 2020 | ● | NAP +3.8 m | 1970 – 2150 | 1970 – 2040 | 1997 - 2150 |



## 2.1 Flood hazard: historical extreme water levels and future projections

Climate change and system-scale interventions, such as the construction of the Maeslant barrier, significantly influence flood
probabilities in the study area. To model these probabilities, we employed Hydra-NL (v.2.8.2), an open-source model widely
used in the Netherlands for deriving hydraulic loads, particularly extreme water levels, to assess and design flood defences
(Ministerie van Infrastructuur en Milieu, 2016). In this study, Hydra-NL was used to derive water level frequency lines for
both historical (1970-2020) and future (2020-2150) scenarios at eight locations along the Nieuwe Waterweg (Fig. 2). Water
level frequency lines represent the magnitude and likelihood of water levels at specific locations. Our methodology
incorporated computations from the 2D-hydrodynamic model WAQUA, along with probabilistic information related to input
variables, including sea water level statistics derived from empirical measurements. The hydrodynamic computations used in
this study are included via the Dutch governmental WBI2017-database, which currently serves as the primary database for
dike assessments and design (Agtersloot and Paarlberg, 2016). This database encompasses hydrodynamic computations for
9,750 combinations of discrete stochastic variables, including wind direction, wind speed, river discharge of the rivers
Nederrijn-Lek, Rhine and Meuse, and the operational status (open/closed) of the Maeslant barrier. Both the historical and
future water level frequency lines were computed using the same WBI2017-database, which means that the impact of sea level
rise on extreme water levels is considered only via the probabilistic analysis, not in the underlying physical combinations.

To calculate historical water level frequency lines (1970-2020), we incorporated past local sea level rise trends: 1.8 mm/year
for 1970-1990 and 2.9 mm/year for 1990-2020, as derived from the Sea Level Rise Monitor 2022 (Deltares, 2023). The impact
of the Maeslant barrier's construction was accounted for by applying a failure probability of 1 per closure request in Hydra-
NL for simulations prior to 1997, effectively removing its influence on extreme water levels before its construction year. While
river discharge distributions of the Rhine are important in calculating extreme water levels in other areas in the Rhine-Meuse
Estuary (system overview in Fig 2a), especially with expected increases in peak discharges that are included in climate
scenarios, their impact on Rotterdam is relatively limited. This is because sea level, including storm surges, has a more
significant impact on water levels in Rotterdam compared to river discharge.

For future water level frequency lines (2020-2150), we adopted the KNMI'23 climate change scenarios (Dorland et al., 2023).
These scenarios, building upon global IPCC scenarios but projecting further into the future, provide a more comprehensive
representation of uncertainty for long-term decision-making and far-future risk management. The rationale behind adopting
the KNMI scenarios is to be methodologically consistent with the IPCC and as such provide a widely accepted and actionable
common projection for climate change in the Netherlands, while at the same time using the most accurate local projections for
Rotterdam. The KNMI'23 scenarios are categorized into low-emission (RCP2.6) and high-emission (RCP8.5) scenarios,
aligning with those used in the Delta Programme, a national initiative evaluating and developing strategies to protect the
Netherlands from flooding, ensure water availability, and enhance the country's resilience to water-related challenges by 2050
(Deltaprogramma, 2023).



In this study, the high-emission scenario serves as point of departure. To illustrate the uncertainty in the timing of sea level rise, we link results from both high- and low-emission scenarios. For instance, the 0.27 m sea level rise projected for 2050 in the high-emission scenario is anticipated by 2058 in the low-emission scenario. This approach is reflected in the double x-axis used throughout the study. Return periods for computing extreme water levels range from 10 to 1000 years.

## 2.2 Exposure: current building stock and future projections

To assess exposure, we utilized the open-source BAG dataset to obtain building footprints of residential assets in Rotterdam's unembanked areas (Kadaster, 2022). For about 9 million addresses in the Netherlands, the BAG database includes several attributes, such as the function of the property (e.g. residential), the surface area of the specific object and the construction year. In the analysis, only residential assets are included. These footprints were combined with elevation data from the Digital Terrain Model of the Netherlands (AHN3, 0.5-meter resolution raster) to assign elevation values to each housing unit. By assigning specific elevation values to each housing unit, we can accurately assess their potential exposure given various climate scenarios It is important to note that the assigned elevation represents the ground level elevation surrounding dwellings and does not account for variations in individual housing unit floor levels. To determine the construction year and locations of planned real estate developments for upcoming decades, we incorporated urban development data from the municipality of Rotterdam, encompassing all significant urban planning projects (Gemeente Rotterdam, 2023). The municipal data includes the number of housing units, their scheduled completion dates, and their respective neighborhoods. As this data lacked elevation information, we developed three scenarios for the elevation of planned housing units:

1. Reference Scenario. All planned housing units are assigned an elevation corresponding to the current design flood elevation of NAP +3.6 m (Amsterdam Ordnance Datum).
2. Increased Design Flood Elevation Scenario: An elevated design flood elevation of NAP +3.8 m is applied.
3. No Design Flood Elevation Scenario: This approach assumes urban densification without raising ground levels. We derived a Gaussian distribution based on existing neighbourhood elevations and used this to sample elevations for planned housing units, maintaining the current elevation distribution per neighbourhood.

**The Design Flood Elevation is one the important components of the long-term flood adaptation strategy for the unenembanked areas of Rotterdam (Sect. 3). Evaluation of various elevations for new residential developments enables assessment of flood elevation policy effectiveness, particularly in light of urban growth and climate change impacts. Although Rotterdam applies more flood reduction measures in its unembanked areas, this research only includes two**





**measures: the design flood elevation policy and the implementation of the Maeslant barrier. 2.3 Combining hazard and exposure**

To quantify exposure, we calculated the difference between water levels at given return periods and the ground level elevation

of each housing unit. A housing unit is considered exposed when this difference exceeds a threshold value of 0.25 m, corresponding to the average height of doorsteps in the study area (Veerbeek et al., 2010).



## 3 Case study: Rotterdam's unembanked areas

The Rhine-Meuse delta in the Netherlands is a highly urbanized area vulnerable to climate change impacts. Adapting to these
challenges is crucial (Van Alphen et al., 2022; De Bruijn et al., 2022; Haasnoot et al., 2020). Our study aims to provide accurate
information on how both physical and societal factors influence flood risk at the neighbourhood level (Fig. 2a). The focus is
on unembanked areas—those located on the river side of primary flood defences, as defined by the Water Act. Unembanked
areas are generally not protected by flood defences and rely on their higher elevation to maintain acceptable flood risk levels.

Rotterdam's flood risk policy for unembanked areas currently recommends raising the ground level of new building lots to
withstand 1000-year flood events under the low KNMI'14 climate scenario (Gemeente Rotterdam, 2021). The design flood
elevation policy aims reduce exposure of new urban development to increasing water levels by raising the ground levels. As
can be seen in Fig. 2d, many unembanked housing units have been constructed on building lots situated below the NAP+ 3.5
m elevation. Given that these unembanked areas generally aren't protected by flood defences, the ground level of new building
lot becomes the dominant factor in mitigating future flood exposure. As the recently released KNMI'23 scenarios project higher
1000-year flood event water levels than the previous scenarios, the municipality plans to revise the current elevation. With the
current design flood elevation policy, new buildings must be elevated up to 1 meter above existing ground levels. Contrarily,
existing urban unembanked areas lack additional regulations to mitigate flood impacts, leaving homeowners responsible for
flood damage and preventive measures (Duijn & van Buuren, 2017).


Focusing solely on achieving specific ground levels would result in a disjointed urban landscape with varying elevations. And
would increase costs for both private investors and the municipality. Consequently, Rotterdam is exploring alternative flood
risk management strategies as addition to the Design Flood Elevation policy. These include not only raising grounds but also
implementing additional measures including sea walls, elevated boulevards, tidal parks, temporary or mobile flood barriers,
and innovative local flood-proofing.. This approach aligns with Rotterdam's broader efforts, like those of other major delta
cities, to proactively address climate change consequences (Ward et al., 2013).





**Figure 2: (a) Rhine-Meuse water system with the North Sea on the west and incoming rivers on the east. (b) Rotterdam's unembanked neighbourhoods and primary flood defences. (c) Water levels by return period, averaged over all hydraulic locations indicated in Fig. 2b. (d) Elevation distribution by neighbourhoods, shown as box plots (median, interquartile range, 5th/95th percentiles); Tarwewijk and Nieuw Mathenesse are excluded because of a limited building stock.**



### 3.1 Extreme water levels between 1970 - 2150

Extreme water levels in Rotterdam depend on the combination of sea level, river discharge, wind speed, wind direction and whether the Maeslant barrier is open or closed. Over time, extreme water levels have increased due to rising sea levels. The extreme water levels in Rotterdam are primarily influenced by sea level rather than river discharge, which sets them apart from other locations in the Rhine-Meuse estuary such as Dordrecht.

The extreme water levels, following from probabilistic calculations, are presented in Fig. 2c. Generally, the extreme water levels show an upward trend, attributed to rising sea levels. This trend assumes no system-scale adaptations, such as increased storage capacity or improvement of the closure reliability of the Maeslant barrier (Mooyaart et al., 2022). From 1996 to 1997, extreme water levels decreased due to the construction of the Maeslant barrier, ranging from a reduction of 0.06 m (10-year events) to 0.71 m (1000-year events). For 1000-year events, the extreme water level in 2150 (RCP8.5) is similar to the water level prior to the construction of the Maeslant barrier. Under the low emission scenario (RCP2.6), for 1000-year events, the extreme water level in 2150 remains 0.48 m lower than 1996 levels. For 100-year events, the extreme water level in 2090 (RCP8.5) equals the water level 1996 level and exceeds the 1996 level by 0.21 m in 2150 (RCP8.5). The mitigating effect of the Maeslant barrier for 10-year events is less because the closing criterium is around the same water level of NAP +3.0 m. The Maeslant barrier is intentionally left open for the majority of 10-year events, and as such, is not designed to mitigate 240 extreme water levels within this specific range. Hence, the extreme water level for 10-year events in 2150 is expected to increase by 0.33 and 0.58 m when compared to the 1996 levels, under the RCP2.6 and RCP8.5 scenarios respectively.

### 3.2 Urban development between 1970 – 2050

The unembanked areas of Rotterdam have been characterized by rapid urban development over the past decades. About 75% of the current building stock in the unembanked areas (n=25,500) is realized after 1980. Furthermore, within the existing 245 building stock, approximately 85% of the housing units are part of multiple housing units within the same building, indicating that the assigned ground level elevation can differ from the actual elevation of the housing unit. The development in the Stadshavens (hereafter: CityPorts) Rotterdam are the largest inner-city development in Europe after London Gateway, covering an area of 1600 ha located on both banks of the river Meuse (Daamen & Vries, 2013; Frantzeskaki et al., 2014). The CityPorts areas are the only port areas situated inside Rotterdam's highway rim, which explains why these are favoured for urban 250 development.

Relocation of harbour activities, toward the sea to accommodate large cargo ships, has opened up spaces in unembanked areas, providing opportunities for new urban (re)developments. Until 2040, Rotterdam is planning to construct 50,000 houses, of which more than 22,600 are located in unembanked areas (Rotterdams Weerwoord, 2022, Gemeente Rotterdam 2023).




The elevation of current unembanked housing units varies across different neighbourhoods (Fig. 2c), which holds implications for designing long-term flood risk adaptation strategies. Findings show that both the mean elevation and width of the distribution differ among these neighbourhoods. These distributions result in varying increases of exposure rates for marginal increases in water levels. Specifically, neighbourhoods such as Heijplaat, Nieuwe Werk, and Noordereiland exhibit median

elevation values below 3.3 meters. When considering all unembanked neighbourhoods collectively, approximately 45% of the existing building stock possesses ground level elevations that fall below the current design flood elevation of NAP +3.6 m. Note that this assessment pertains not to the actual height of the housing units but rather to the elevation of the ground on which they are situated.

## 4 Results

The results are presented in four sections. Section 4.1 presents the flood exposure in Rotterdam's unembanked areas from 1970 to 2150, examining the combined effects of sea level rise, urban development, and the construction of the Maeslant barrier. Section 4.2 then highlights temporal variations in exposure, demonstrating how flood risk has changed over time. Subsequently, Sect. 4.3 presents an attribution analysis, examining and comparing the individual effects of sea level rise, urban development, and the Maeslant barrier construction on flood exposure. The analysis concludes in Sect. 4.4 with a sensitivity

assessment, evaluating how flood exposure responds to changes in design flood elevation levels.

### 4.1 Flood exposure: urban development, sea level rise and adaptation combined

An analysis of the collective influence of sea level rise, urban development, and the Maeslant barrier construction on the exposure of unembanked housing units in Rotterdam was conducted for the past 50 years and projected 130 years (Fig. 3). Without adaptation measures, unembanked areas in Rotterdam are expected to experience increased flood exposure,

particularly in the 22nd century. The magnitude of this increase varies across different return periods. For 10-year events, the number of houses exposed to flooding is projected to increase from 400 in 1970 to 800 in 2020, and further to 5,700 by 2150. 100-year flood events show a different pattern: exposure decreases from 3,200 houses in 1970 to 1986, in 2020, followed by an increase to 9,100 in 2150 (RCP8.5). Similarly, for 1000-year events, exposure initially decreases from 5,200 houses in 1970 to 4,700 in 2020, before rising significantly to 39,400 in 2150 (RCP8.5).


Under the RCP8.5 scenario, these projections translate to a 7-fold increase in exposure for 10-year events, a 5-fold increase for 100-year events by 2150 and an 8-fold increase for 1000-year events, compared to 2020 levels. The RCP2.6 scenario projects less severe increases: a 3-fold increase for 10-year and 100-year events, and a 2-fold increase for 1000-year events by 2150.






The operationalization of the Maeslant barrier in 1997 marked mitigation in exposure, evidenced by a decrease in extreme water levels (Fig. 2b) and exposed houses (Fig. 3). Specifically, the number of exposed houses decreased by approximately 100 houses (10%) for 10-year events, 2,600 houses (62%) for 100-year events, and 7,300 houses (67%) for 1000-year events. The barrier's impact for 10-year events, compared to the other return periods, is limited because extreme water levels during

such events do not always surpass the barrier's closure threshold. Consequently, there are instances in which the Maeslant barrier remains open, even though doing so results in flood exposure for certain properties.

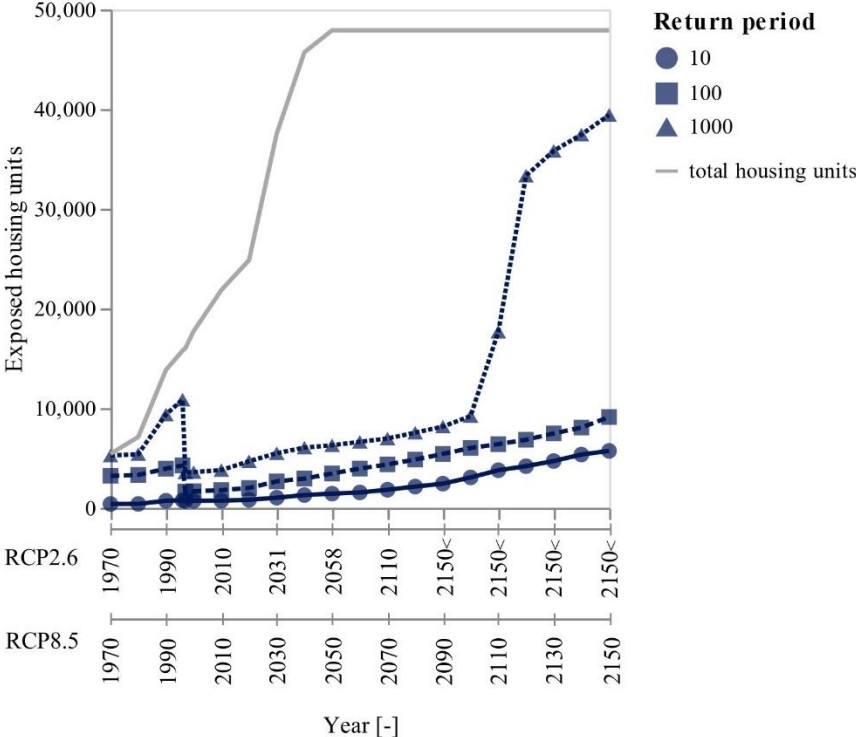

**Figure 3: Number of exposed houses for Rotterdam's unembanked areas for 10-, 100- and 1000-year events between 1970 and 2150, for the low KNMI'23 (RCP2.6) and high KNMI' 23 (RCP8.5) scenarios.**




## 4.2 Spatial distribution of exposure

Rotterdam's 14 unembanked neighbourhoods show significant variations in flood susceptibility, largely due to differences in housing unit elevations (Fig. 2c). To illustrate these exposure variations, we analyzed the proportion of exposed houses in each neighbourhood over time (Fig. 4). Percentages fluctuate as the total number of houses changes (due to urban development)
and water levels shift (sea level rise and construction of the Maeslant barrier). This analysis reveals significant disparities in the percentage of vulnerable houses within unembanked areas of Rotterdam, emphasizing the need for neighbourhoods-specific adaptation strategies.

For 10-year flood events, the range of exposed houses across neighbourhoods is relatively narrow. In 1970, Kop van Zuid –
Entrepot was the most vulnerable neighbourhoods with 48% (100 houses) exposed, followed by Feyenoord with 24% (200 houses). Other neighbourhoods had less than 5% exposure. By 2020, exposure generally decreased, with the Esch neighbourhood highest at 17% and 9 out of 14 neighbourhoods showing no exposure. Projections for 2150 under the RCP8.5 scenario indicate exposure reaching up to 58% in Noordereiland, while under the RCP2.6 scenario, maximum exposure in any neighbourhood is limited to 23%.


The 100-year flood events show greater variability in exposure rates. In 1970, all neighbourhoods, except for the 6 neighbourhoods with few or no houses in 1970, had over 43% exposure. By 2020, exposure decreased across all neighbourhoods, reaching up to 23% in Nieuwe Werk. Projections for 2150 under the RCP8.5 scenario suggest a wide range, from 0% exposure in Nieuw Mathenesse to 76% in Heijplaat.


For 1000-year events, the variability in exposure rates between neighbourhoods decreases, with many of them approaching 100%. Nearly all houses were at risk in 1970, dropping to between 11% and 77% by 2020. However, 2150 projections under RCP8.5 show a return to near-total exposure in most areas.

Interestingly, while extreme water levels have increased linearly since 1997 (Fig. 2b), the growth in exposed houses follows a less predictable pattern. This is due to neighbourhood-specific elevation profiles. These characteristics influence both the timing (e.g., when 50% of the housing units become exposed for a specific return period) and the temporal progression (e.g., the duration required for a marginal increase in exposure). For example, Nieuwe Werk, with its lower average elevation and less variability, shows a rapid increase to 100% exposure. In contrast, Kop van Zuid, with higher and more varied elevations,
shows a more gradual increase in risk. These findings highlight that one-size-fits-all solutions, like uniform waterfront heightening, won't be equally effective across Rotterdam. Each neighbourhood requires a unique approach to flood risk management, based on its specific elevation profile and urban development patterns.



**Figure 4: Percentage of homes exposed in neighbourhoods for 10-, 100-, and 1000-year flood events across multiple years, a) 1970, b) 2020, c) 2150< (RCP2.6) & 2100 (RCP8.5) and d|) 2150< (RCP2.6) & 2150 (RCP8.5).**



### 4.3 Disentangling urban development, sea level rise and adaptation

Flood exposure in Rotterdam's unembanked areas is influenced by an interplay of urban development, sea level rise and adaptation measures. Comprehending the interaction among these factors is key in effective planning for the city's growth and designing flood risk adaptation strategies. We assess these factors by examining their individual impacts to inform effective city planning and flood risk management strategies.

To assess the sensitivity of exposure to future urban development, we conducted a comparative analysis of scenarios with and without post-2020 urban development. Our findings reveal significant disparities primarily when extreme water levels surpass NAP +3.85 meters, exceeding the design flood elevation of NAP +3.6 m (which includes a 0.25 m threshold). This threshold is only exceeded during 1000-year flood events (Fig. 2c). Hence, the comparative subplots show that stopping urban development after 2020 could ultimately prevent exposure to 1000-year events of all 22,600 housing units planned for construction. For lower return periods like 100-year events, the impact of urban development is minimal until after 2150 under the RCP8.5 climate scenario. Interestingly, we observed some exposure differences between 2020 and 2100 across all return periods. As we assigned actual ground levels, rather than design elevation levels, to the 3,600 housing units built between 2020 and 2023, these variations are attributable to our arbitrary selection of the year 2020 after which urban development is stopped. About 55% of the housing units constructed after 2020 have ground levels below the design flood elevation of NAP +3.6 m.

The impact of sea level rise after 2020 on flood exposure is also assessed by comparing exposure with and without post-2020 sea level rise (Fig. 5). Until 2100 (RCP8.5), exposure rates increase marginally and consistently. After 2100 (RCP8.5), the exposure rates for 100-year events rise more sharply, primarily because most housing elevations, among others due to the design flood elevation, are close to the corresponding extreme water level (Fig. 2b). The impact of sea level rise after 2020 is larger than the impact urban development. Without post-2020 sea level rise, exposure rates are expected to be 6 times lower for 10-year flood events, 4 times lower for 100-year flood events, and 7 times lower for 1000-year flood events.

The Maeslant barrier plays a crucial role in mitigating flood exposure in Rotterdam. Without this barrier, water levels in Rotterdam would directly reflect sea level changes, significantly increasing flood risk. Our analysis reveals that the Maeslant barrier has effectively reduced flood exposure, with its impact most pronounced for 1000-year events (Fig. 5). By 2150, under the high emission scenario (RCP8.5), the absence of the Maeslant barrier would result in substantially higher exposure rates. Specifically, exposure would be 7 times higher for 10-year events, 5 times higher for 100-year events, and 1.2 times higher for 1000-year events. Even under the low emission scenario (RCP2.6), the barrier's absence would lead to 3, 7, and 6 times higher exposure rates for 10-year, 100-year, and 1000-year events respectively. The Maeslant barrier's effectiveness in reducing exposure rates outweighs the combined increases caused by urban development and sea level rise.





While the Maeslant barrier significantly postpones the onset of increased flood exposure due to climate change, it does not
offer permanent protection. This limitation is evident in the projected increase in exposure for 1000-year events, demonstrating
that the barrier cannot indefinitely counteract the long-term flood risk escalation driven by sea level rise.



**Figure 5: Influence of urban development, sea level rise and construction of the Maeslant barrier on the number of exposed houses.**
**The top three figures show the absolute number of exposed houses, the lower three figures are relative to the reference. The reference**
**scenario incorporates the combined effects of sea level rise, urban development, and the Maeslant barrier construction.**





### 4.4 Impact design flood elevation on exposure

Elevating new housing can be an effective strategy to reduce their vulnerability to future flood hazards. This section analyzes the impact of this adaptation measure on exposure rates by comparing two other strategies: one without a design flood elevation

requirement and another with an elevated design flood elevation of NAP +3.8 m. These two strategies are contrasted with the previous analyses in this study, which considered existing building stock at current ground level elevations and new housing units (built after 2020) meeting a design flood elevation of NAP +3.6 m.

The absence of a design flood elevation requirement generally leads to higher exposure rates across most return periods and

reference years compared to scenarios with a design flood elevation (Fig. 6). Projections for the year 2150 under the RCP8.5 scenario illustrate this trend. Without a design flood elevation, exposure rates for 10-year events are expected to increase by 54%, while 100-year events could see a 71% increase. Interestingly, 1000-year events present a contrasting picture, with an anticipated 24% reduction in exposure rates. The decrease in exposure for 1000-year events in 2110, when comparing the no-design elevation scenario to the reference, is due to the elevation distribution in the no-design flood elevation scenario. In this

case, some newly built houses are assigned elevations exceeding the NAP +3.6 m design flood elevation, as the current elevation distribution includes areas with higher ground levels (Fig. 2d).

Raising the design flood elevation to NAP +3.8 m yields modest reductions in exposure rates compared to the current NAP +3.6 m standard. For 10-year and 100-year events, this elevation increase doesn't significantly impact exposure rates, as water

levels typically remain below both elevation thresholds. The benefits become apparent for 1000-year events, where the higher design flood elevation results in reduced exposure rates after 2100 under the RCP8.5 scenario. The 0.2-meter elevation increase effectively postpones the rise in exposure rates by approximately 30 years, providing additional time for neighbourhoods to implement further adaptation measures. While the impact on exposure rates may seem modest, the elevated design flood elevation of NAP +3.8 m offers other benefits. It decreases inundation depths during flood events, which directly translates to

reduced property damage and lower recovery costs.





**Figure 6: Influence of design flood elevation on the number of exposed houses. The top three figures show the absolute number of exposed houses when the current elevation distribution per neighbourhoods is applied for new housing units, design flood elevation NAP +3.8 m gives the situation when building lots are raised to NAP +3.8 m. The lower three figures are relative to the reference.**



## 5 Discussion

This study developed a framework to evaluate the combined impacts of urban development, sea level rise and adaptation on flood exposure, applying it to Rotterdam's unembanked areas. The framework developed in this study is relevant for delta cities worldwide, while maintaining adaptability to local contexts. The core methodological components can be transferred in three ways. First, hazard and exposure data can be adapted using locally available models and exposure datasets. While our study used Dutch datasets (BAG & AHN), the framework can accommodate other data sources like building footprints, elevation data and local urban development plans. Second, the temporal analysis for attributing flood exposure changes to specific factors (e.g. urban development) offers a universal approach for understanding risk drivers in coastal cities. This attribution analysis is particularly relevant for (port) cities experiencing similar transitions in their unembanked areas. Third, the neighborhood-level analysis for researching spatial variations in exposure can be replicated in other urban contexts, informing targeted flood adaptation strategies.

Our approach combines various models and data sources, introducing inherent uncertainties that must be considered when interpreting results. A primary constraint arises from the temporal disparity in data projections: future urban development data extends only to 2040, while climate scenarios project to 2150. This restricts our ability to assess long-term exposure in unembanked areas and attribute changes to urban development beyond 2040.

Another limitation concerns the elevation data for future urban development. Our study assumes that the elevation of newly built housing units complies with the design flood elevation. We uniformly assign ground level elevation to new housing units, including apartments in high-rise structures. Consequently, our exposure assessment does not necessarily reflect direct exposure of individual housing units but rather indicates instances where the lower floor is exposed.

The third limitation of our study lies in its omission of explicit considerations for local flood-proofing measures, such as seawalls. To illustrate, most quay walls in Feijenoord have elevations exceeding NAP +3.15 m, higher than some housing units' elevations, with the lowest at approximately NAP +2.5 m. As a result, our model may overestimate exposure in areas where such protective structures exist, as extreme water levels exceeding housing unit elevations in Feijenoord may not lead to actual exposure in reality. To partially address minor flood protection measures, our approach uses a uniform 0.25 m threshold, representing elevated entrances and door guards. This simplification could be refined in future studies by incorporating neighbourhood or building-specific thresholds that more accurately reflect real-world conditions.

Our study's probabilistic calculations are based on the 2017 hydraulic computations database, which is the standard for hydraulic and dike assessments in the Netherlands' water system. A fourth limitation of our approach is that sea level rise is only factored into the probabilistic calculations, not the underlying hydraulic computations. This approach has several implications. For instance, the Mean Tide Level (MTL) remains unchanged in our hydrodynamic computations, potentially leading to underestimated extreme water levels, particularly for lower return periods. Furthermore, our model does not account




for significant infrastructural changes such as the construction of the 2nd Maasvlakte (reclaimed land), alterations to the Nieuwe Waterweg canal over time, or potential future adjustments to the Maeslant barrier's operation. The last few decades have demonstrated the profound impact such interventions have on estuarine hydrodynamics. Ideally, new databases with hydrodynamic computations including sea level rise would be derived. This need becomes particularly apparent for scenarios where sea level rise exceeds 1 meter, as the combined effects of rising seas and altered coastal infrastructure could significantly reshape flood risk patterns in ways our current framework cannot fully capture.


Our findings on increasing flood exposure align with previous studies focused on the unembanked areas of the Rhine-Meuse estuary, though with some notable differences. De Moel et al. (2014) projected a doubling of expected annual damage by 2100 from a 2014 baseline of 40 million euros. In contrast, Veerbeek et al. (2010) reported a significantly lower initial expected annual damage of 0.16 million euros but predicted a more rapid increase, with expected annual damage doubling by 2050 and quadrupling by 2100. This large difference in baseline expected annual damage values between the two studies needs further investigation and may be attributed to varying methodologies, data sources, or scope of analysis. Our exposure projections for 2050, based on the KNMI'23 high emission scenario, align with the findings of Veerbeek et al. (2010), differing by less than 5%. Our 2100 projections are approximately 20% lower than those of Veerbeek et al. (2010), primarily due to the different sea level rise scenarios used. Our study uses the KNMI'23 high emission scenario with an expected sea level rise of 0.87 m, while Veerbeek et al. (2010) employed the Veerman scenario with 1.3 m sea level rise.

As direction for further research we propose to expand the model with both physical and (social) vulnerability assessments in our flood risk analysis. Integrating these factors would allow for a more comprehensive evaluation of flood risk and inform the design of more effective long-term adaptation strategies. This is particularly crucial for Rotterdam's unembanked areas, where about 20% of the population is classified as highly socially vulnerable, compared to only 6% in embanked areas (Koks et al., 2014). It is important to note that urban development in these unembanked areas is not solely a risk factor, but also presents significant opportunities for urban improvement. For example, the Kop van Feijenoord project aims to create thousands of new homes while simultaneously improving flood defences and public spaces. Such development can serve as a catalyst for strengthening neighbourhoods and enhancing connectivity between Rotterdam's northern and southern districts.

Also, damage modelling is proposed to be included to make estimates for the expected annual damage in the present and future. Given the heterogeneity in housing units, ranging from low-lying historic buildings to high-rise buildings, integrating sophisticated damage modelling could provide more accurate risk assessments. Using advanced damage curves, such as the framework of Schlumberger et al. (2023) for Venice, are particularly valuable when assessing older buildings. To model future flood risk, especially when considering planning horizons over 50 years, asset value is key for projections. In 465 2050, flood risk is expected to double in Europe solely based on the increase in value of exposed buildings (Steinhausen et al., 2022). At the same time, increasing exposure rates or flood events can make flood risk more salient, which could lead to a revaluation of properties located in unembanked areas (Caloia et al., 2023; Jansen, 2023).





## 6 Conclusion

This study has developed and applied a comprehensive flood exposure assessment framework to Rotterdam's unembanked
areas, providing insights into the interplay of urban development, sea level rise, and adaptation strategies. The framework is
applied to Rotterdam's unembanked areas, which have a building stock of 25 500 housing units and urban development plans
for an additional 22 600 in the next two decades.

Without additional adaptation, the unembanked areas in Rotterdam are expected to experience increases in flood exposure,
especially in the 22nd century under the high emission scenario of the KNMI'23 (RCP8.5). Under the RCP8.5 scenario, we
project a 7-fold increase in exposure for 10-year events, a 5-fold increase for 100-year events by 2150 and an 8-fold increase
for 1000-year events, compared to 2020 levels. The RCP2.6 scenario projects less severe increases: a 3-fold increase for 10-
year and 100-year events, and a 2-fold increase for 1000-year events by 2150. The study highlights the critical role of the
Maeslant storm surge barrier in mitigating flood risk. Its implementation has approximately halved exposure rates for 100-
year and 1000-year events compared to 1996 levels. Our analysis shows that without this barrier, exposure rates would increase
dramatically, mirroring increases in local water levels due to sea level rise.

Spatial variability in exposure rates across the 14 unembanked neighborhoods emphasizes the importance of localized data for
urban planning and neighbourhood-level adaptation strategies. This variability is particularly pronounced for 10- and 100-year
events. By 2150, under the high-emission RCP8.5 scenario, variation in exposure is large: neighbourhoods such as Nieuwe
Werk, Heijplaat, and Noordereiland are projected to face exposure rates exceeding 65% for 100-year flood events. This is
more than three times the aggregated exposure percentage of 19% for all unembanked areas, underscoring the potential for
certain areas to become hotspots of flood risk.

Findings show that the implementation of the Maeslant barrier is dominant in exposure rates, followed by sea level rise and
urban development.

No urban development after 2020 will not have a significant effect on exposure rates before 2100 (RCP8.5) and 2150
(RCP2.6). After 2100 (RCP8.5), we can expect increases in exposure rates for 1000-year events as the extreme water levels
coincide with the design flood elevation for new housing units. This indicates that the current building stock, rather than future
urban development, predominantly dictates short to mid-term adaptation strategies for Rotterdam's unembanked areas.

The hypothetical case of the absence of sea level rise after 2020 would lead to a reduction in exposure rates by 2150
(RCP8.5). Exposure rates are expected to be 6 times lower for 10-year flood events, 4 times lower for 100-year flood events,
and 7 times lower for 1000-year flood events. Especially the impact of sea level rise on lower return periods highlights the
vulnerability of the estuarine unembanked areas.




Without the construction of the Maeslant barrier, extreme water levels would mirror sea levels resulting in significant increases of exposure. By 2150, under the high emission scenario (RCP8.5), the absence of the Maeslant barrier would result in substantially higher exposure rates. Specifically, exposure would be 7 times higher for 10-year events, 5 times higher for 505 100-year events, and 1.2 times higher for 1000-year events. Even under the low emission scenario (RCP2.6), the barrier's absence would lead to 3, 7, and 6 times higher exposure rates for 10-year, 100-year, and 1000-year events respectively. The Maeslant barrier's closure frequency directly impacts flood exposure rates in Rotterdam. With rising sea levels, the barrier is projected to close more often, potentially increasing from once every 10 years to several times annually by 2100. This impacts flood exposure rates and port accessibility. To limit the number of closures, authorities are considering raising the closure 510 water level, which would decrease the barrier's effectiveness and increase exposure rates in unembanked areas, particularly for 10-year and 100-year flood events. This study's framework can assess the impact of such threshold changes on flood exposure in unembanked areas.

Moreover, the study evaluated the efficacy of elevating housing units as a flood mitigation strategy. Under the high-emission 515 scenario (RCP8.5) by 2150, implementing the current design flood elevation (NAP +3.6 m) reduces exposure by 54% for 10-year flood events and 71% for 100-year flood events, compared to the no design elevation strategy. Increasing the elevation to a design flood elevation of NAP+ 3.8 m has no impact on 10-year events and 100-year events within the studied timeframe but offers a temporary reduction in exposure for 1000-year events between 2100 (RCP8.5) and 2130 (RCP8.5). Raising the Maeslant barrier's closure threshold increases flood risk for 10-year and 100-year events, but this can be offset by elevating 520 new building lots. This interplay highlights the need for combining local neighbourhood-specific strategies with regional flood management: the effectiveness of local design elevation policies depends on the barrier's closure threshold, while the impact of adjusting the threshold varies based on neighbourhood elevation levels. While the specific exposure rates and effectiveness of adaptation measures are unique to Rotterdam, the importance of understanding neighborhood-level variations and the interplay between system-scale adaptation measures and local elevation policies are relevant for coastal cities worldwide. In 525 port cities like Houston, Hamburg and Copenhagen, port relocation is creating new urban development opportunities in unembanked areas. These cities face similar challenges in coordinating system-scale infrastructure (like storm surge barriers) with local measures (such as elevated building lots) to manage long-term flood risk under climate change uncertainty.

Looking ahead, the unembanked areas around Rotterdam will become increasingly important. After 2040, unembanked areas of the larger Rotterdam area are expected to have over 100,000 residents, coupled with their significance for industry, port 530 operations, and valuable ecosystems, the stakes are high. The dilemma of whether to permanently close the Rhine-Meuse estuary or keep an open connection during daily conditions for gravitational river discharge underscores the complexity of the challenges ahead. For the national long-term strategy, the first dilemma that may emerge is whether to permanently close the estuaries or to keep them open to facilitate river discharge by gravity. Hence, there is no doubt that Rotterdam's unembanked areas are pivotal in formulating the long-term adaptation strategy of the Netherlands, and vice versa.



*Code and data availability.* The python notebook used for the analysis and visualizations can be retrieved from https://data.4tu.nl/datasets/d1291401-708a-4d48-9d95-8259cfd987d2 (Oerlemans, 2024). Flood hazard data, such as extreme water levels for return periods and locations used in this study is included as well. Urban development plans on a project level are not included but can be obtained upon request from the municipality of Rotterdam.

*Author contributions.* Conceptualization: CO, MB, TR and MK; data curation: CO; formal analysis: CO and MB; methodology: CO; flood hazard modelling: CO; exposure modelling: CO; writing – original draft: CO; writing – review and editing: MB, TR and MK; supervision: MB and MK; funding acquisition: MB and MK. All authors have read and agreed to the published version of the manuscript.

*Competing interests.* The authors declare that they have no conflict of interest.

*Acknowledgements.* This research is part of the RED&BLUE research program (Real Estate Development & Building in Low Urban Environments, NWA.1389.20.224). We thank the municipality of Rotterdam, Dook Ligthart, Vera Konings and Liselotte Mesu, for providing valuable insights and their willingness to share urban development plans. We acknowledge the use of Claude 3.5 Sonnet (Anthropic, https://claude.ai/) for proofreading the final draft.

*Financial support.* This publication is part of the project RED&BLUE, Real Estate Development & Building in Low Urban Environments with project number NWA.1389.20.224 of the research program Dutch Research Agenda (NWA)] which is financed by the Dutch Research Council (NWO).



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
