# Peer review of "Flood exposure in Rotterdam's unembanked areas from 1970 to 2150: sensitivities to urban development, sea level rise and adaptation"

_EGUsphere, 2024_

## Author Comment (AC1)

Dear anonymous referee,

Thank you for your constructive comments which have helped improve our manuscript. We highly appreciate your time, effort and support. The point by point reply below includes our responses to your comments and how we adjusted the manuscript accordingly.

**Introduction**

*"The study conducts an exposure assessment for unembanked areas in Rotterdam for the years 1970 to 2150 accounting for different extreme water levels. Building exposure is estimated for different scenario combinations and neighborhoods to identify the contribution of sea level rise, urban development and adaptation (i.e. construction of the Maeslant barrier and elevating new buildings) to the exposure. The study presents a comprehensive framework for local exposure assessments, integrating future development of building infrastructure and adaptation. The manuscript is written clearly and the text is supported by insightful figures. Though the idea of the exposure assessment does not present an entirely new approach and the findings are sometimes relatively obvious, the study combines many different contributing factors to future exposure and the framework is interesting for further local studies. I have some general and also more specific comments that I believe could increase the scientific and presentation quality of the manuscript. I hope the authors consider the comments useful."*

Response: We are glad that you find the manuscript relevant and well written. While exposure analysis itself is not new, our contribution indeed lies in combining regional sea level rise projections with detailed local urban development data across both historical analysis and future projections. The scientific added value is the combination of these analyses to reach novel insights on developing flood risk adaptation strategies by accounting for urban development changes over time and neighborhood-level differences in exposure that would be lost in national or regional flood risk assessments.

**General comments**

*"The discussion and conclusion could benefit from some restructuring. Right now, the discussion mostly presents the limitations of the study, whereas the conclusion discusses the implications of the results. This should also be part of the discussion and the conclusion should then only highlight on the main findings."*

Response: We have restructured the conclusion and discussion as suggested. The discussion section now addresses both limitations and implications, while the conclusion contains only the main findings.

*"The Methods describing the calculation of the flood hazards are not described sufficiently for readers to be able to reproduce them. How does the Hydra-NL model incorporate the results from the hydrodynamic model to calculate water level frequency lines at different locations? Which hydrodynamic model combinations have been used from the database? It would be helpful to add a short explanation to the manuscript."*

Response: Thank you for your sharp observation. We have expanded our explanation of Hydra-NL for calculating water level frequency lines to assist the reproducibility of the calculations. Hydra-NL uses the entire database of hydrodynamic computations, weighing the contribution of each stochastic variable combination based on exceedance probability, sea level rise, and Maeslant barrier operation. Detailed breakdowns of contributions of hydrodynamic combinations are available in the supplementary material (https://data.4tu.nl/datasets/d1291401-708a-4d48-9d95-8259cfd987d2). For example: rotterdam_unembanked\5_hydranl\computations.zip\WBI2017_Benedenrijn_17-2_v04\017-02_0018_9_NM_km0995\Berekeningen\m000_exMLK\uitvoer.html. In our manuscript, we have added references to these sources as well.

*"In the results, exposure between neighborhoods is expressed and compared based on the relative share of exposed housing units in each neighborhood. You state for example that Kop van Zuid-Entrepot is the most vulnerable neighborhood with 48% of the housings being exposed. However, in absolute numbers these "only" refer to 100 exposed houses. In this context for example Feyenoord would be more exposed (200 houses). From a city planning perspective, I think it might be more relevant to know about the absolute exposure in order to identify hotspots and prioritize actions in neighborhoods where many houses and thus people are affected. The absolute numbers might be especially relevant for the 1000-year event, where 100 % of the houses are exposed for almost all neighborhoods. This could also be picked upon in the discussion to reflect, that evaluating exposure is also a matter of perspective and scale."*

Response: We acknowledge that from a city planner's perspective, both the absolute as well as the relative exposure are valuable. While we previously highlighted absolute exposure in Section 4.1 and relative exposure for the neighborhoods in Section 4.2, we have now also included absolute exposure figures in Section 4.2 to better represent the contributions of different neighborhoods.

*"Throughout the manuscript the use of the terms exposure, risk and vulnerability is inconsistent and often erroneous. Since this is an exposure analysis, I would suggest to stick to this term and be careful when using the terms risk and especially vulnerability."*

Response: Thank you for your observation. The terms exposure, risk and vulnerability each have their own meaning and purpose. However, we understand that this may raise confusion for readers from different domains. Hence, we have carefully revised our use of terminology throughout the manuscript, maintaining focus on 'exposure' and limiting references to 'risk' and 'vulnerability' only where specifically applicable. For situations related to the increase in exposure, we now use the term susceptibility rather than vulnerability. We also explicitly added the definitions of these terms where we introduced them.

**Specific comments:**

*"Page 3, line 74-86: I find it unusual to refer to the findings already in the introduction. I would suggest to only describe the four-step process here without any interpretation of the results yet."*

Response: We have chosen to share some of the findings in the introduction as they provide readers with a clearer expectation of the article content, which we believe increases readability for a broader audience. In the revised manuscript, we have framed these findings as research questions rather than explicit results.

*"Page 4, line 91-101: The first paragraph of the Methods and data section reads more like part of the introduction. Perhaps, try to reduce this paragraph and/or include it in the introduction."*

Response: We have shortened the first paragraph of the method section and moved the relevant content to the introduction.

*"Page 4, line 109: Maybe mention that altering the design flood elevation is a scenario accounting for adaptation/accommodation."*

Response: We have clarified that adjusting the design flood elevation is part of an adaptation/accommodation flood risk adaptation strategy.

*"Page 7, line 178: Why did the authors choose a scenario which involves elevating the design flood level by just 20 cm? Are there concrete actions/plans considering this as a new design height? To me it doesn't sound like too much of a difference to the reference scenario and I am wondering why the authors didn't choose a higher design level? Also, the authors could consider to elaborate other adaptation options in their discussion, like e.g. household-level adaptation, which might eliminate exposure in cases of lower water levels."*

Response: Indeed, the 20 cm elevation increment reflects concrete, though not yet approved, proposals from Rotterdam municipality. This might seems a minor adaptation measure, but it would already involve costs approximating tens of millions of euros while significantly reducing/delaying exposure. Household-level adaptation, and the trade-off between household-level adaptation and system-scale adaptation, is interesting and is one of the topics to include in next studies. For the current manuscript we focused on investigating on system scale adaptation measures (storm surge barriers and land-raising of new building lots).

*"Page 8, line 189-191: To quantify exposure, the authors calculate the difference between water levels and the elevation of the housing unit. It appears that they are not accounting for hydrological connectivity here. This means housings behind structures higher than the water level are eventually not flooded. It would be good to mention this as a limitation. A similar issue is mentioned in the discussion about the missing seawall information."*

Response: Thank you. We have expanded our discussion on missing seawall information to include how buildings affect hydrological connectivity. To be brief, there are seawalls that prevent water from initially flooding particular sections. However, these structures are relatively low (compared to the water depths), and our analysis and discussions with the municipality revealed that there are no internal structures that would limit flood propagation within neighborhoods.

*"Page 15, line 341-343: "Hence the comparative subplots show that stopping urban development after 2020 could ultimately prevent exposure to 1000-year events of all 22,600 housing units planned for construction." I find this finding a bit too obvious. Perhaps there is no need to write this."*

Response: We have retained this sentence as it also serves as a reference point for our subsequent exposure findings related to lower return periods (line 343-347).

*"Page 20, line 421-427: Is the height information of the flood-proofing measures not reflected in the DTM? Referring to my earlier comment, if elevation information from the DTM is accurate for protections measures it might be valuable to assess exposed areas based on hydrological connectivity rather than comparing the water level to each housing elevation."*

Response: While 2D modeling, or accounting for hydrological connectivity in another way, would improve accuracy for current conditions, we learned that the benefits are marginal for the entire period between 1970–2150 as uncertainties in land use changes and digital terrain models across this period are dominant. Of course, uncertainties regarding hydrological connectivity over time can be mitigated as well, but that would require many additional calculations and calculation time.

*"Page 21, line 441-450: The authors compare their estimates to the results of other exposure studies. Though from what they write it sounds like these studies assess exposure as monetary value. How is the comparison of these studies with the estimates of relative exposure of housing units from this study justified?"*

Response: Thank for bringing this up. ,We agree that it would be better to compare our exposure figures to exposure figures from other studies. However, exposure figures are not explicitly mentioned in these studies as they only included Expected Annual Damage (EAD). The outcomes of these studies are still considered relevant as they elaborate on the same greater Rotterdam region.

Hence, we have included these studies and compared the results, but now with an explicit acknowledgment that comparing exposure results (our study) to risk results – from de Moel et al. (2014) and Veerbeek et al. (2010) - comes with limitations.

*"Page 23, line 525: Is there a reference for port relocation in Hamburg? In the introduction this is only mentioned for Houston, Copenhagen and Rotterdam."*
Response: We have added Hamburg to the introduction, making it consistent with the discussion section where Hamburg is already mentioned.

*"Temporal line plots: I think it could be beneficial to explain the time line differences of the x-axis scales in each line plot caption in case readers are just looking into the figures."*
Response: We have added brief explanations of the time scale differences in each figure caption.

**Technical comments:**
We have implemented almost all suggested technical corrections:

*"Page 1, line 15-18: "Temporal variations in exposure rates are attributed […] and urban development." The two sentences mean the same and one of them could be removed here."*
Response: we understand your confusion. The sentences are not meant to have the same content, as the second sentence provides information about the magnitude of the attribution. We revised the sentences to better clarify their intended meaning.

*"Page 5, Figure 1: Maybe write in the "exposure"-box that the data for urban development plans only reaches until 2040 to make sure it does not cover the full period of investigation."*
Response: We have updated Figure 1 to clarify the urban development data horizon.

*"Page 5, line 120: It should be Table 1 here, not Table 2."*
Respond. We apologize and corrected the table reference on page 5.

*"Page 7, line 161: I suggest to mention all tested return periods here instead of writing "from 10 to 1000 years"."*
Response: We listed all return periods explicitly.

*"Page 7, line 169: "These footprints were combined with elevation data from the Digital Terrain Model of the Netherlands (AHN3, 0.5-meter resolution raster) [to assign elevation values to each housing unit]. By assigning specific elevation values to each housing unit, we can accurately assess their potential exposure given various climate scenarios." I would suggest the authors to remove the words in [...] to avoid repetition and add a full stop at the end of the second sentence."*
Response: We removed repetitive phrasing about elevation data.

*"Page 7, line 183: "The Design Flood Elevation is one the important components …" Look into the grammar here."*
Response: We fixed the grammar in this sentence.

*"Page 7, line 183 – page 8, line 188: Paragraph is formatted like the subsequent heading."*
Response: We fixed the paragraph formatting issues.

*"Chapter 3: Chapter 3 appears to be a bit detached from the rest of the text. I would propose to put chapter "3 Case Study: Rotterdam's unembanked areas" at the beginning of the Methods section as*

*a description of the study site. Chapter 3.1. and 3.2 can be included in the Results, as they describe the findings of your flood hazard calculation and data processing."*

Response: We intend to have a distinct chapter to introduce the Rotterdam case. This approach allows us to provide not only data but also background on why Rotterdam is developing in a particular manner and how the current water system and policies came into place. We have included both urban development data and extreme water level data in this chapter so that we can focus on the results of the exposure analysis in the results section.

*"Page 9, line 203: Should it be "have been constructed on building lots situated below the NAP+3.6 m elevation", as this is the design flood elevation?"*

Response: In this case, it does not specifically refer to the design flood elevation of 3.6 m+NAP. Therefore, we have kept the original reference of 3.5 m+NAP and clarified the text to make this more explicit.

*"Page 11, line 242: "3.2 Urban development between 1970-2050" Should it rather be 2040 as this is the time span of the urban development plans?"*

Response: We revised the urban development timeline from "1970-2050" to "1970-2040".

*"Page 17, Figure 5: The reference scenario should be mentioned in the legend."*

Response. The reference is included in the vertical axis which represents the scenario minus reference.

Thank you again for your constructive and sharp comments. We hope our responses adequately address your comments.

Kind regards,
Cees Oerlemans on behalf of the co-authors.

---

## Author Response (AR1)

Date: 29-05-2025

Dear Margreth Keiler and anonymous referees,

We are pleased to submit our revised manuscript titled "Flood exposure in Rotterdam's unembanked areas from 1970 to 2150: sensitivities to urban development, sea level rise and adaptation."

We thank the editor and both referees for their constructive feedback, which has helped to improve the overall quality of our manuscript. This document provides our point-by-point responses to all comments received during the review process.

List of point-by-point replies:

- Editor: pre-print submission
- Referee #1 comment #1
- Referee #1 comment #2
- Referee #2 comment #1
- Editor: final submission

We look forward to your consideration of our revised submission.

Kind regards,

Cees Oerlemans

On behalf of the authors Martine van den Boomen, Ties Rijcken and Matthijs Kok

**Submission for pre-print posting (editor)**

Date posted: 17-12-2024 Data author response: 20-12-2024

This comment was posted by the editor upon article submission. Since the manuscript underwent further revisions after this comment was made, some of our replies have been updated and differ from the original author response.

**Editor comment**

Please add a brief review on studies addressing to disentangle the risk drivers hazard, exposure, vulnerability and the retroperspective modelling approach, e.g. examples form riverine flooding in addition to coastal flooding.

**Author reply**

We have incorporated this feedback by explicitly referencing studies that focus on disentangling the risk drivers - hazard, exposure, and vulnerability - and relating their findings to the context of our research.

Specifically, we added the following paragraph:

"Climate change intensifies these flood risks through more frequent storms, rising sea levels and higher river peak discharges (Calafat et al., 2022; Oppenheimer et al., 2019). Flood risk is commonly defined as a function of hazard, exposure and vulnerability. Several studies have disentangled the key drivers of changes in flood risk across different spatial scales. At continental and global scales, historical analyses from 1950 to 2020 show that the most important drivers of flood impacts in Europe have been exposure growth and vulnerability decline (Paprotny et al., 2024). Steinhausen et al. (2022) analyzed the independent and combined influence of exposure change and climate scenarios on future flood risk in Europe finding that exposure change has a greater influence than climate-driven hazard changes in the near to mid-term future (up to 2085). Global analyses reveal distinct regional patterns. While flood impacts in African countries are mainly driven by climate change, in growing Southeast Asian economies (Indus, Yangtze, and Mekong basins), rapid urban growth dominates over climate change effects (Winsemius et al., 2016).

These large-scale studies highlight general trends, but designing effective local flood reduction strategies requires understanding risk drivers at finer spatial resolutions. At the

regional and local scale, Koks et al. (2014) provided a framework to jointly assess flood hazard, exposure, and social vulnerability, demonstrating that including detailed regional information on flood risk drivers is crucial for developing effective flood reduction strategies. Local system dynamics can be complex: Schlögl et al. (2021) demonstrated that interactions within coupled natural and socioeconomic systems maintain stable flood risk outcomes even as hazard events grow. Insights into these multi-scale risk drivers underscore that adaptation - through structural measures (sea walls, levees), nature-based solutions (mangrove restoration, wetlands), building-level interventions or urban development policies - can significantly reduce future flood losses even under lower global warming scenarios (Magnan et al., 2022; Oppenheimer et al., 2019; Aerts, 2018; Song et al., 2017)."

- b) You highlighted transferability of your approach to worldwide deltas in the introduction. However, I miss the answer of the discussion and conclusion. Which part of your research design or methods can be transferred to other regions. It seems that your manuscript is very focused to your case study. Please stress this aspect in your manuscript to show the scientific added value of your study.
- Upon reflection and rereading the discussion and conclusion sections, we acknowledge that the transferability of our findings was indeed understated. In response, we have substantially expanded the discussion section to: (1) provide broader perspective on result interpretation, (2) explicitly address framework limitations and their implications for transferability, and (3) strengthen comparisons with existing literature to better demonstrate our scientific contribution.
- c) Exposure is a main component in your study, thus please also extend the information on exposure dynamics as addressed introduction by more detailed information in the methods, e.g. exposure dynamic and influencing factors, assumption for evolution until 2040 beside the design flood elevation.
- The entire methods section has been revised to address these points and improve reproducibility.
- d) The readers of NHESS are not all familiar with the Netherlands and therefore I ask you to add more information on the adaptation strategies and which one are considered in your study. Explain in more detail e.g. the design flood elevation.

We adjusted the manuscript to provide a more comprehensive introduction to the research context, explicitly mentioning that the scope of the study focuses exclusively on two flood reduction measures: 1) the design flood elevation policy for new housing developments, and 2) the construction of the Maeslant barrier. Moreover, together with point c), we elaborated on the background of the adaptation strategy of Rotterdam's

unembanked areas, including more in-depth context for the design flood elevation policy.

**Referee #1 comment #1**

Date posted: 11-03-2025 Date author response: 15-04-2025

This is Reviewer #1's first comment. Note that because the manuscript underwent additional revisions after this comment was submitted, some of our replies below reflect these later changes and differ from our initial response.

**Referee comment Author reply General comments** "The discussion and conclusion could benefit We have restructured the conclusion and from some restructuring. Right now, the discussion as suggested. The discussion section discussion mostly presents the limitations of now addresses both limitations and implications, the study, whereas the conclusion discusses while the conclusion contains only the main the implications of the results. This should findings. also be part of the discussion and the conclusion should then only highlight on the main findings." "The Methods describing the calculation of Thank you for your sharp observation. We have the flood hazards are not described expanded our explanation of Hydra-NL for sufficiently for readers to be able to calculating water level frequency lines to assist reproduce them. How does the Hydra-NL with the reproducibility of the calculations. Hydramodel incorporate the results from the NL uses the entire database of hydrodynamic hydrodynamic model to calculate water computations, weighing the contribution of each level frequency lines at different locations? stochastic variable combination based on Which hydrodynamic model combinations exceedance probability, sea level rise, and have been used from the database? It Maeslant barrier operation. Detailed breakdowns would be helpful to add a short explanation of the contributions of hydrodynamic to the manuscript." combinations are available in the supplementary material (https://data.4tu.nl/datasets/d1291401-708a-4d48-9d95-8259cfd987d2). For example: rotterdam\_unembanked\5\_hydranl\computations. zip\WBI2017\_Benedenrijn\_17-2\_v04\017-02\_0018\_9\_NM\_km0995\Berekeningen\m000\_ex MLK\uitvoer.html. In our manuscript, we have added references to these sources as well. We acknowledge that from a city planner's *In the results, exposure between* perspective, both the absolute as well as the neighborhoods is expressed and compared based on the relative share of exposed relative exposure are valuable. While we previously highlighted absolute exposure in housing units in each neighborhood. You state for example that Kop van Zuid-Section 4.1 and relative exposure for the Entrepot is the most vulnerable neighborhoods in Section 4.2, we have now also neighborhood with 48% of the housings included absolute exposure figures in Section 4.2**

being exposed. However, in absolute numbers these "only" refer to 100 exposed houses. In this context for example Feyenoord would be more exposed (200 houses). From a city planning perspective, I think it might be more relevant to know about the absolute exposure in order to identify hotspots and prioritize actions in neighborhoods where many houses and thus people are affected. The absolute numbers might be especially relevant for the 1000-year event, where 100 % of the houses are exposed for almost all neighborhoods. This could also be picked upon in the discussion to reflect, that evaluating exposure is also a matter of perspective and scale."

to better represent the contributions of different neighborhoods.

"Throughout the manuscript the use of the terms exposure, risk and vulnerability is inconsistent and often erroneous. Since this is an exposure analysis, I would suggest to stick to this term and be careful when using the terms risk and especially vulnerability."

Thank you for your observation. The terms exposure, risk and vulnerability each have their own meaning and purpose. However, we understand that this may raise confusion for readers from different domains. Hence, we have carefully revised our use of terminology throughout the manuscript, maintaining focus on 'exposure' and limiting references to 'risk' and 'vulnerability' only where specifically applicable. For situations related to the increase in exposure, we now use the term susceptibility rather than vulnerability. We also explicitly added the definitions of these terms where we introduced them.

**Specific comments**

Page 3, line 74-86: I find it unusual to refer to the findings already in the introduction. I would suggest to only describe the fourstep process here without any interpretation of the results yet. We have chosen to share some of the findings in the introduction as they provide readers with a clearer expectation of the article content, which we believe increases readability for a broader audience. In the revised manuscript, we have framed these findings as research questions rather than explicit results.

"Page 4, line 91-101: The first paragraph of the Methods and data section reads more like part of the introduction. Perhaps, try to reduce this paragraph and/or include it in the introduction." We have shortened the first paragraph of the method section and moved the relevant content to the introduction.

"Page 4, line 109: Maybe mention that altering the design flood elevation is a scenario accounting for adaptation/accommodation."

We have clarified that adjusting the design flood elevation is part of an adaptation/accommodation flood risk adaptation strategy. "Page 7, line 178: Why did the authors choose a scenario which involves elevating the design flood level by just 20 cm? Are there concrete actions/plans considering this as a new design height? To me it doesn't sound like too much of a difference to the reference scenario and I am wondering why the authors didn't choose a higher design level? Also, the authors could consider to elaborate other adaptation options in their discussion, like e.g. household-level adaptation, which might eliminate exposure in cases of lower water levels."

Indeed, the 20 cm elevation increment reflects concrete, though not yet approved, proposals from the Rotterdam municipality. This might seem like a minor adaptation measure, but it would already involve costs approximating tens of millions of euros while significantly reducing/delaying exposure. Household-level adaptation, and the trade-off between household-level adaptation and system-scale adaptation, is interesting and is one of the topics to include in next studies. For the current manuscript we focused on investigating on system scale adaptation measures (storm surge barriers and land-raising of new building lots).

Page 8, line 189-191: To quantify exposure, the authors calculate the difference between water levels and the elevation of the housing unit. It appears that they are not accounting for hydrological connectivity here. This means housings behind structures higher than the water level are eventually not flooded. It would be good to mention this as a limitation. A similar issue is mentioned in the discussion about the missing seawall information.

Thank you. We have expanded our discussion on missing seawall information to include how buildings affect hydrological connectivity. To be brief, there are seawalls that prevent water from initially flooding particular sections. However, these structures are relatively low (compared to the water depths), and our analysis and discussions with the municipality revealed that there are no internal structures that would limit flood propagation within neighborhoods.

Page 15, line 341-343: "Hence the comparative subplots show that stopping urban development after 2020 could ultimately prevent exposure to 1000-year events of all 22,600 housing units planned for construction." I find this finding a bit too obvious. Perhaps there is no need to write this.

We have retained this sentence as it also serves as a reference point for our subsequent exposure findings related to lower return periods.

Page 20, line 421-427: Is the height information of the flood-proofing measures not reflected in the DTM? Referring to my earlier comment, if elevation information from the DTM is accurate for protections measures it might be valuable to assess exposed areas based on hydrological connectivity rather than comparing the water level to each housing elevation.

While 2D modeling, or accounting for hydrological connectivity in another way, would improve accuracy for current conditions, we learned that the benefits are marginal for the entire period between 1970–2150 as uncertainties in land use changes and digital terrain models across this period are dominant. Of course, uncertainties regarding hydrological connectivity over time can be mitigated as well, but that would require many additional calculations and calculation time.

Page 21, line 441-450: The authors compare their estimates to the results of other exposure studies. Though from what they write it sounds like these studies assess exposure as monetary value. How is the comparison of these studies with the

Thank you for bringing this up. We agree that it would be better to compare our exposure figures to exposure figures from other studies. However, exposure figures are not explicitly mentioned in all studies as they included Expected Annual Damage (EAD). The outcomes of these studies

| estimates of relative exposure of housing units from this study justified?" | are still considered relevant as they elaborate on the same greater Rotterdam region.  Hence, we have included these studies and compared the results, but now with an explicit acknowledgment that comparing exposure results (our study) to risk results – from de Moel et al. (2014) and Veerbeek et al. (2010) - comes with limitations. |
|-----------------------------------------------------------------------------|----------------------------------------------------------------------------------------------------------------------------------------------------------------------------------------------------------------------------------------------------------------------------------------------------------------------------------------------|
| Page 23, line 525: Is there a reference for                                 | We have added Hamburg to the introduction,                                                                                                                                                                                                                                                                                                   |
| port relocation in Hamburg? In the                                          | making it consistent with the discussion section                                                                                                                                                                                                                                                                                             |
| introduction this is only mentioned for                                     | where Hamburg is already mentioned.                                                                                                                                                                                                                                                                                                          |
| Houston, Copenhagen and Rotterdam.                                          |                                                                                                                                                                                                                                                                                                                                              |
| "Temporal line plots: I think it could be                                   | We have added brief explanations of the time                                                                                                                                                                                                                                                                                                 |
| beneficial to explain the time line                                         | scale differences in each figure caption.                                                                                                                                                                                                                                                                                                    |
| differences of the x-axis scales in each line                               | scale differences in each figure caption.                                                                                                                                                                                                                                                                                                    |
|                                                                             |                                                                                                                                                                                                                                                                                                                                              |
| plot caption in case readers are just looking                               |                                                                                                                                                                                                                                                                                                                                              |
| into the figures."                                                          |                                                                                                                                                                                                                                                                                                                                              |
| Technical comments                                                          |                                                                                                                                                                                                                                                                                                                                              |
|                                                                             |                                                                                                                                                                                                                                                                                                                                              |
| Page 1, line 15-18: "Temporal variations in                                 | We understand your confusion. The sentences are                                                                                                                                                                                                                                                                                              |
| exposure rates are attributed [] and urban                                  | not meant to have the same content,                                                                                                                                                                                                                                                                                                          |
| development." The two sentences mean the                                    | as the second sentence provides information                                                                                                                                                                                                                                                                                                  |
| same and one of them could be removed                                       | about the magnitude of the attribution. We                                                                                                                                                                                                                                                                                                   |
| here.                                                                       | revised the sentences to better clarify their                                                                                                                                                                                                                                                                                                |
|                                                                             | intended meaning.                                                                                                                                                                                                                                                                                                                            |
| Page 5, Figure 1: Maybe write in the                                        | We have updated Figure 1 to clarify the urban                                                                                                                                                                                                                                                                                                |
| "exposure"-box that the data for urban                                      | development data horizon. As the manuscript is                                                                                                                                                                                                                                                                                               |
| development plans only reaches until 2040                                   | restructured, this is currently Figure 2.                                                                                                                                                                                                                                                                                                    |
| to make sure it does not cover the full                                     |                                                                                                                                                                                                                                                                                                                                              |
| period of investigation.                                                    |                                                                                                                                                                                                                                                                                                                                              |
| "Page 5, line 120: It should be Table 1                                     | We apologize and corrected the table reference.                                                                                                                                                                                                                                                                                              |
| here, not Table 2."                                                         | , -                                                                                                                                                                                                                                                                                                                                          |
| "Page 7, line 161: I suggest to mention all                                 | We listed all return periods explicitly.                                                                                                                                                                                                                                                                                                     |
| tested return periods here instead of writing                               | , , ,                                                                                                                                                                                                                                                                                                                                        |
| "from 10 to 1000 years"."                                                   |                                                                                                                                                                                                                                                                                                                                              |
| "Page 7, line 169: "These footprints were                                   | We removed repetitive phrasing about elevation                                                                                                                                                                                                                                                                                               |
| combined with elevation data from the                                       | data.                                                                                                                                                                                                                                                                                                                                        |
| Digital Terrain Model of the Netherlands                                    |                                                                                                                                                                                                                                                                                                                                              |
| (AHN3, 0.5-meter resolution raster) [to                                     |                                                                                                                                                                                                                                                                                                                                              |
| assign elevation values to each housing                                     |                                                                                                                                                                                                                                                                                                                                              |
| _                                                                           |                                                                                                                                                                                                                                                                                                                                              |
| unit]. By assigning specific elevation values                               |                                                                                                                                                                                                                                                                                                                                              |
| to each housing unit, we can accurately                                     |                                                                                                                                                                                                                                                                                                                                              |
| assess their potential exposure given                                       |                                                                                                                                                                                                                                                                                                                                              |
| various climate scenarios." I would suggest                                 |                                                                                                                                                                                                                                                                                                                                              |
| the authors to remove the words in [] to                                    |                                                                                                                                                                                                                                                                                                                                              |
| avoid repetition and add a full stop at the                                 |                                                                                                                                                                                                                                                                                                                                              |
| end of the second sentence."                                                |                                                                                                                                                                                                                                                                                                                                              |

| Page 7, line 183: "The Design Flood           | We fixed the grammar in this sentence.               |
|-----------------------------------------------|------------------------------------------------------|
| Elevation is one the important components     |                                                      |
| " Look into the grammar here."                |                                                      |
| "Page 7, line 183 - page 8, line 188:         | We fixed the paragraph formatting issues.            |
| Paragraph is formatted like the subsequent    |                                                      |
| heading."                                     |                                                      |
| Chapter 3: Chapter 3 appears to be a bit      | We have restructured the manuscript by moving        |
| detached from the rest of the text. I would   | the case study introduction from section 3 to the    |
| propose to put chapter "3 Case Study:         | methods section (2.1). Additionally, the trend       |
| Rotterdam's unembanked areas" at the          | descriptions for both extreme water level analysis   |
| beginning of the Methods section as           | and urban development have been relocated to         |
| a description of the study site. Chapter 3.1. | the first section of the results (3.1). These        |
| and 3.2 can be included in the Results, as    | changes have significantly improved the narrative    |
| they describe the findings of your flood      | of the study.                                        |
| hazard calculation and data processing."      |                                                      |
| "Page 9, line 203: Should it be "have been    | In this case, it does not specifically refer to the  |
| constructed on building lots situated below   | design flood elevation of 3.6 m+NAP. Therefore,      |
| the NAP+3.6 m elevation", as this is the      | we have kept the original reference of 3.5 m+NAP     |
| design flood elevation?"                      | and clarified the text to make this more explicit.   |
| "Page 11, line 242: "3.2 Urban development    | We revised the urban development timeline from       |
| between 1970-2050" Should it rather be        | "1970-2050" to "1970-2040".                          |
| 2040 as this is the time span of the urban    |                                                      |
| development plans?"                           |                                                      |
| "Page 17, Figure 5: The reference scenario    | The reference is included in the vertical axis which |
| should be mentioned in the legend."           | represents the scenario minus reference.             |

**Referee #1 comment #2**

Date posted: 15-04-2025 Date author response: 23-04-2025

This comment contained follow-up questions and requests for clarification based on the author's response. Since the manuscript underwent further revisions after this comment was made, some of our replies have been updated and differ from the original author response.

| Editor comment                                  | Author reply                                      |
|-------------------------------------------------|---------------------------------------------------|
| You say that "For situations related to the     | Indeed, if we refer to exposure or an increase in |
| increase in exposure, we now use the term       | exposure we phrase it a such. We only use         |
| susceptibility rather than vulnerability.". Why | "susceptibility" when discussing neighborhoods    |
| would you interchange these terms. To my        | that may experience increased exposure, always    |
| understanding susceptbility is different to     | making this relationship with exposure explicit   |
| exposure. If you want to refer to an increase   | to avoid ambiguity.                               |
| in exposure I think you should also phrase it   |                                                   |
| as such.                                        |                                                   |
| 2) Regarding the comment Page 15, line 341-     | The exposure of planned housing depends on        |
| 343: What I thought was quite obvious about     | multiple factors: elevation of planned units,     |
| this sentence is that of course the planned     | flood return period, sea level rise scenario      |
| units will not be exposed in the future if      | (high/low) and reference year. While unbuilt      |

urban development was stopped completly in 2020. And I guess that this applies irrelevant of the return period. I am not sure if my remark on this came across properly before.

houses cannot be exposed, the difference in exposure between scenarios with/without urban development varies with return period. For larger return periods, this difference equals the total number of planned units. For lower return periods, fewer housing units may be affected. We will clarify this distinction to prevent confusion.

**Referee #2 comment #1**

Data posted: 07-04-2025 Date author response: 23-04-2025

The following is the first comment provided by Reviewer #2.

**Referee comment**

**Author reply**

**General comments**

"This study performs an exposure assessment for the unembanked areas of Rotterdam from 1970 to 2150, considering different extreme water levels (historical sea level rise, NAP + 3.6m and future sea level rise resulting from RCP2.6 and RCP8.5). The exposure of buildings is assessed under different combinations of these flood scenarios and planned urban development, and under different adaptation options (construction of the Maeslant barrier and elevation of new buildings). Although large parts of the manuscript are clearly written, there are some concerns about the overall structure of the manuscript. In addition, the reference to other studies seems a bit tricky, as the studies the authors refer to have been carried out at a regional scale, which makes it difficult to compare the results. Nevertheless, the study is within the scope of the journal as it effectively brings together several factors influencing flood exposure. The framework presented could be valuable for similar local studies. I have a number of general comments, listed below, which may help the authors to improve the scientific rigour and overall

structure of the manuscript."

We are glad that you find the framework valuable for similar local studies. We have addressed your suggestions by reviewing the references to others studies, restructuring several sections of the manuscript and including specific aspects as detailed below.

**Introduction**

"The authors deduce the research gap by referring to previous studies on exposure and changing flood risk resulting from climate change. These studies, however, have been carried out on a global to European scale, and limit the implications for local studies. Specifically, European studies based on assessments using the NUTS3 level cannot easily compared to the local, building plot level. As such, while the studies of Steinhausen et al. (2022), Praprotny et al. (2024) and others are valuable, a direct conclusion for the identified research gaps needs further elaboration (comparison is a matter of scale as the resolution of the data and therefore the preciseness is spatially variable, see for example Karagiorgos et al. (2024)). Possible other local studies in this field of research include those of Schlögl et al. (2021) focusing on similar topics, and results may be compared with the results of the present Rotterdam study."

The referred to studies are indeed on another scale. The reason why we referred to these studies is two-fold: 1) to study whether the patterns observed at larger scales – where exposure growth often dominates - hazard growth also applies on a local level and specifically our case-study in Rotterdam and 2) to highlight that while there are more larger scale studies than local studies in the literature, local mechanisms are important for developing local adaptations strategies. In response to your observation, we have now clearly separated these points in the manuscript for better clarity. We appreciate your suggestions regarding Karagiorgos et al. (2024) and Schlögl et al. (2021) - these works provide valuable local-scale perspectives that we have incorporated into our revised introduction.

Furthermore, the authors are kindly advised to carefully check their use of terms – as this study explicitly addresses dynamics of flood hazard and exposure taking into account adaptation measures reducing exposure, the reference to the vulnerability term should be omitted (also in other sections of the manuscript) – also increasing exposure will not necessarily result in increasing vulnerability (cited from page 2, line 27), see the underlying UNDRR definitions (https://www.undrr.org/drrglossary/terminology).

Indeed, we have reviewed the use of the terms hazard, exposure and vulnerability. Concretely, regarding line 27, we have deleted the word vulnerability. In addition we went through the entire manuscript and revised where necessary.

The authors may also consider to make their focus on residential buildings much clearer already here, as building types different from residential ones are excluded here – specifically also with respect to the material provided on page 3, lines 60ff. ("...risk estimates would benefit from a more detailed consideration of objects and land-use categories"), but also with respect to the concluding sentences provided in lines 71f.

We understand your point. In response we have revised the manuscript to clarify our focus on residential buildings much earlier in the text. To be short, we used detailed object-based residential data (BAG) which provides higher precision than the coarser land-use categories that poorly represent the characteristics of Rotterdam's unembanked areas. This approach, while more precise for housing, comes at the expense of not including other categories - businesses, nature, industry - in the exposure analysis.

"The authors may wish to further adapt the four points presented as workflow for the study, in some of the aspects results are already summarised which typically should not be presented in the intro section. Also, "various urban development strategies" (point 4) explored by only "varying the design flood elevation for new development" seems a bit overwhelming as in fact, only elevation of ground was considered (and not other local flood protection such as e.g. local structural measures) – hence, the list could benefit from a thorough re-wording and formulation."

Thanks for your observations. We revised the wording in the revised version of the manuscript with the purpose of enhancing clarity and being more specific about what is being researched and why.

**Methods and data**

"The authors are kindly asked to check the wording here, and to delete the references to "vulnerability" (see my comments above, this should be made clear in the introduction)."

We agree. In line with other comments regarding the terminology, we have systematically revised the wording throughout the entire manuscript.

Flood hazard: the data used to compute the flood hazard are not entirely clear to me, the authors are kindly asked to provide more details to allow the potential readers an evaluation and reproduction (see also the comment of reviewer 1). Authors may consider at least an Appendix here so that details can be given with sufficient details."

Thank you. We fully agree that our research needs to be reproducible and this has been our intention from the start as our approach is not meant to be limited to our case study but transferable to other regions. Apparently, we did not yet sufficiently succeed in bringing this reproducibility stand out. We extended our manuscript and our supplementary material on this aspect. In the manuscript we improved our elaboration on technical specifications of the hydrodynamic model and the data used in this study. Additionally, we expanded the supplementary material and included references from the main text to the supplementary material.

Exposure: please check the formatting of the last paragraph on page 7. Moreover, I am a bit confused about the exposure scenarios as I understood that an elevation of 3.6 and 3.8 m was considered as adaptation measure. How does such a high elevation influence the flood hazard? Does this simply imply that all the new constructions are flood safe then, and if yes, this is a bit obvious as this is a universally valid assumption? The authors could clarify their methods in this direction a bit, also with respect to other adaptation

Thank you for your observation about the formatting, we have corrected this. We have clarified our implementation of adaptation measures, particularly regarding the flood design elevation levels. The proposed levels of 3.6 and 3.8 m+NAP provide protection against more frequent events, but will not result in full protection for new building stock (which would indeed result in obvious exposure observations).

probably prescribed to homeowners in the Netherlands."

**Case study selection**

Here the authors already present results from the historical analysis which should be moved to the results section (specifically sections 3.1 and 3.2). Moreover, the statement on page 9, lines 195f. are not clear to me as the study does not build on "societal factors" influencing flood risk on the neighbourhood level. Authors are kindly asked to re-write. Lines 211ff: This paragraph weakens the method used in the study, and such constraints or limitations should be moved to the discussion section."

Following your suggestion, we have restructured our case study chapter and moved parts of the historical analysis (former sections 3.1 and 3.2) to the results section. The phrase 'societal factors' is indeed used in an ambiguous manner. Our intention was to refer to urban development plans so we replaced 'societal factors' by 'urban development plans'. Regarding the text of line 211 and beyond; these are not meant to refer to our methods, but rather to the adaptation strategy adopted by the municipality of Rotterdam. We have clarified this distinction in the revised manuscript.

**Results section**

I am bit confused about the fact that the authors rely on a relative comparison of exposure only, and do not provide information on the absolute values as they strongly differ between the different neighbourhoods."

We acknowledge that for a full urban planning perspective both the relative as well as the absolute exposure is relevant. While we previously highlighted absolute exposure in Section 4.1 and relative exposure for the neighborhoods in Section 4.2, we have now also included absolute exposure figures in Section 4.2 to better represent the contributions of different neighborhoods.

"Figures: resolution should be higher to enhance accessibility and readability."

We have enhanced the figures by formatting the lay-out to improve readability.

**Comments of the editor**

Date posted: 03-05-2025 Date author response: 26-05-2025

This comment was placed after the replies to the referees, indicating that the manuscript needed major revisions.

| Editor comment                              | Author reply                                   |
|---------------------------------------------|------------------------------------------------|
| As you know, two reviewers have now         | Our responses above address each comment       |
| provided detailed reviews, which you have   | while restructuring the document to provide a  |
| replied in thoughtful detail to. Both       | clearer narrative and highlight the scientific |
| reviewers recommended major revisions,      | contributions.                                 |
| and therefore I would like to invite you to |                                                |
| submit a revised version of your            |                                                |
| manuscript. Please consider also the        |                                                |
| comments I addressed before the             |                                                |

| manuscript was transferred to NHESS-D, some of them were also addressed again by the reviewers. It is important to use the correct terms, improve the overall structure and clearly highlight the scientific addedvalue of your study.                                                                         |                                                          |
|----------------------------------------------------------------------------------------------------------------------------------------------------------------------------------------------------------------------------------------------------------------------------------------------------------------|----------------------------------------------------------|
| Would you please also provide an 'author's reply' to the reviewers (feel free to use the same words that you used in what you have already uploaded). Please can you also include a track changes document between the old manuscript and the new one (you can include this as part of your 'author's reply'). | Yes, of course. This document is included in the portal. |